# Screening of Molecular Targets of Action of Atractylodin in Cholangiocarcinoma by Applying Proteomic and Metabolomic Approaches

**DOI:** 10.3390/metabo9110260

**Published:** 2019-11-01

**Authors:** Kanawut Kotawong, Wanna Chaijaroenkul, Sittiruk Roytrakul, Narumon Phaonakrop, Kesara Na-Bangchang

**Affiliations:** 1Graduate Program in Bioclinical Sciences, Chulabhorn International College of Medicine, Thammasat University, Paholyothin Road, Klonglung, Pathumthani 12120, Thailand; wizard___T_T@hotmail.com (K.K.); wn_ap39@yahoo.com (W.C.); 2Center of Excellence in Pharmacology and Molecular Biology of Malaria and Cholangiocarcinoma, Thammasat University, Paholyothin Road, Klonglung, Pathumthani 12120, Thailand; 3Center for Genetic Engineering and Biotechnology (BIOTEC), National Science and Technology Development Agency, Pathumthani 12120, Thailand; sittiruk@biotec.or.th (S.R.); narumon.pha@biotec.or.th (N.P.)

**Keywords:** cholangiocarcinoma, atractylodin, proteomics, metabolomics, LC-MS/MS, targets of action

## Abstract

Cholangiocarcinoma (CCA) is cancer of the bile duct and the highest incidence of CCA in the world is reported in Thailand. Our previous in vitro and in vivo studies identified *Atractylodes lancea* (Thunb) D.C. as a promising candidate for CCA treatment. The present study aimed to examine the molecular targets of action of atractylodin, the bioactive compound isolated from *A. lancea*, in CCA cell line by applying proteomic and metabolomic approaches. Intra- and extracellular proteins and metabolites were identified by LC-MS/MS following exposure of CL-6, the CCA cell line, to atractylodin for 24 and 48 h. Analysis of the protein functions and pathways involved was performed using a Venn diagram, PANTHER, and STITCH software. Analysis of the metabolite functions and pathways involved, including the correlation between proteins and metabolites identified was performed using MetaboAnalyst software. Results suggested the involvement of atractylodin in various cell biology processes. These include the cell cycle, apoptosis, DNA repair, immune response regulation, wound healing, blood vessel development, pyrimidine metabolism, the citrate cycle, purine metabolism, arginine and proline metabolism, glyoxylate and dicarboxylate metabolism, the pentose phosphate pathway, and fatty acid biosynthesis. Therefore, it was proposed that the action of atractylodin may involve the destruction of the DNA of cancer cells, leading to cell cycle arrest and cell apoptosis.

## 1. Introduction

Cancer is uncontrolled cell proliferation, and it is the leading cause of mortality worldwide. Liver and colon cancers, including cholangiocarcinoma (CCA) are cancer types with high mortality and incidental rates. CCA is cancer of the bile duct and the highest incidence in the world is reported in Thailand. It is classified into intrahepatic (iCCA), perihilar (pCCA), and distal (dCCA) CCA based on anatomical position. iCCA develops in liver parenchymal cells, and accounts for 20–25% all CCA cases [1]. The 3 and 5-years survival rates are 30% and 18%, respectively [2]. pCCA originates from the epithelial cells of the hepatic duct and spreads to common hepatic and cystic ducts. The five-year survival rates of pCCA are reported at 11%. dCCA is located in the area from the cystic duct to the ampulla of Vater. The incidence of this cancer type is about 20–30% of all CCA cases. The five-year survival rate of pCCA is about 11%. Gemcitabine, cisplatin, and 5-fluorouracil (5-FU) are currently used to treat CCA, but their clinical efficacy is unsatisfactory. Research and development of alternative medicines, particularly those from natural sources are challenging. Our previous in vitro and in vivo studies identified *Atractylodes lancea* (Thunb) D.C. (AL) as a promising candidate for CCA treatment [3]. AL has long been used as a traditional medicine in China, Japan, and Thailand. The two main bioactive compounds isolated from the AL rhizome are atractylodin (14%), and β-eudesmol (6%) [4]. It has been shown that the inhibitory action of atractylodin on cancer cell growth is through suppression of dendritic cell maturation [5], the delay of gastric emptying [6], and regulation of interleukin-6 in HMC-1 by blocking NPM-ALK activation [7]. The present study aimed to further examine other molecular targets of action of atractylodin in the CCA cell line by applying proteomic and metabolomic approaches.

## 2. Results

### 2.1. Proteomics

A total of 4366 and 4383 proteins were identified in the intra- and extracellular components of the treated (atractylodin) and untreated CL-6 cells after 24 h of incubation. The corresponding numbers of proteins after 48 h incubation were 4391 and 4348 proteins, respectively. The analysis of Venn diagrams suggested that 659 and 507 proteins from the intracellular components were only found in atracylodin-treated and untreated cells after 24 h incubation, respectively. The corresponding number of proteins in the extracellular components after 24 h incubation was 674 and 532 proteins, respectively (Figure 1a,b). These proteins were selected for gene ontology analysis by PANTHER software. Results showed that 585 proteins were involved in biological processes. Most (192 proteins) were involved in cellular processes (GO:0009987), followed by metabolic processes (GO:0008152) (140 proteins), and biological regulation (GO:0065007) (97 proteins) (Figure 2a). Also, 25 proteins were involved in transporter activity, i.e., ATP-binding cassette sub-family G member 8, CSC1-like protein 1, gamma-aminobutyric acid receptor subunit epsilon, and melanoma inhibitory activity protein 2 (Table 1). Analysis of protein-protein interactions of the 585 proteins by STITCH software suggested the involvement of these proteins in various biological processes including the cell cycle, DNA repair, apoptosis, regulation of immune response, wound healing, and blood vessel development (Figure 3a).

For the 48 h incubation, the analysis of the Venn diagrams suggested that 606 and 564 proteins from the intracellular components were only found in atracylodin-treated and untreated cells, respectively. The corresponding number of proteins in the extracellular components after 48 h incubation was 655 and 538 proteins, respectively (Figure 1c,d). Gene ontology analysis by PANTHER showed that 645 proteins were involved in biological processes. Most (192 proteins) were involved in the cellular process (GO:0009987), followed by the metabolic process (GO:0008152) (148 proteins), and biological regulation (GO:0065007) (102 proteins) (Figure 2b). Also, 34 proteins were involved in transporter activity, i.e., adenosine 3’-phospho 5’-phosphosulfate transporter 2, amiloride-sensitive sodium channel subunit gamma, and ATP-binding cassette sub-family A member 12 (Table 1). Analysis of protein-protein interactions of the 645 proteins by STITCH software (Figure 3b), suggested the involvement of these proteins in cell morphogenesis, cell adhesion, cell cycle, DNA repairing, apoptosis, regulation of immune response, and wound healing.

### 2.2. Metabolomics

Following exposure of CL-6 cell to atractylodin for 24 and 48 h, 289 metabolites from both ionization modes (positive and negative) were identified in the intra- and extracellular components by LC-MS/MS. The area under the peak of each metabolite was calculated and compared with the control (untreated cell). Metabolites with a change in the peak area of at least 2-fold were selected for further analysis. For the 24 h exposure, 21, 19, 11, and 16 metabolites were up-regulated intracellular, down-regulated intracellular, up-regulated extracellular, and down-regulated extracellular metabolites, respectively. The metabolite that showed the highest fold-change was the up-regulated intracellular metabolite methyl nicotinamide (8.57-fold), followed by the up-regulated intracellular metabolite inosine (8.05-fold), the down-regulated intracellular metabolite glutathione disulfide (6.84-fold), and the up-regulated intracellular metabolite phenylpyruvate (6.25-fold). Metabolites that showed the lowest fold-change were the up-regulated intracellular metabolite pyridoxine (2.01-fold) and the down-regulated intracellular metabolite UTP (2.01-fold) (Table 2). The top five metabolic pathways identified by MetaboAnalyst software included aspartate metabolism, pyrimidine metabolism, lactose synthesis, ammonia recycling, and glutamate metabolism (Figure 4a). For the 48 h exposure, 58, 11, 8, and 57 metabolites were up-regulated intracellular, down-regulated intracellular, up-regulated extracellular, and down-regulated extracellular metabolites, respectively. The metabolite that showed the highest fold-change was the down-regulated extracellular metabolite glutathione disulfide (65.53-fold), followed by the down-regulated intracellular metabolite glutathione disulfide (61.87-fold), the up-regulated intracellular metabolite taurodeoxycholic acid (22.41-fold), and the up-regulated intracellular metabolite 2-aminooctanoic acid (21.35-fold). The metabolite that showed the lowest fold-change was the up-regulated intracellular metabolite pantothenate (2.00-fold) (Table 3). The top five metabolic pathways identified by MetaboAnalyst software included the citric acid cycle, pyrimidine metabolism, Warburg effect, glycine and serine metabolism, and aspartate metabolism (Figure 4b).

### 2.3. Co-Analysis of Proteomics and Metabolomics

The proteomics and metabolomics information were co-analyzed for a 24 h and 48 h exposure period. Proteins from the intra- and extracellular components that were found only in atractylodin-treated and untreated CL-6 cells, together with the up-regulated and down-regulated metabolites from both intra- and extracellular components following 24 and 48 h incubation were selected for analysis. For the 24 h exposure, results from MetaboAnalyst analysis revealed four proteins that are associated with six metabolites (Figure 5a). The pathways associated with these interactions are pyrimidine metabolism, purine metabolism, and the pentose phosphate pathway. The biological processes that are linked to these pathways are blood coagulation and wound healing (Figure 5a). For the 48 h exposure, results from MetaboAnalyst analysis revealed 29 proteins that are associated with 20 metabolites (Figure 5b). The pathways associated with these interactions are pyrimidine metabolism, the citrate cycle (TCA cycle), taurine and hypotaurine metabolism, biotin metabolism, purine metabolism, arginine and proline metabolism, glyoxylate and dicarboxylate metabolism, the pentose phosphate pathway, fatty acid biosynthesis, and primary bile acid biosynthesis. The biological processes that are linked to these pathways are blood coagulation, inflammatory response, regulation of the immune system process, regulation of cell proliferation, Ras protein signal transduction, and DNA repairing (Figure 5b).

## 3. Discussion

In this study, proteomic and metabolomic approaches were applied for the identification of the possible molecular targets of action of atractylodin in cholangiocarcinoma. The proteomics results showed that most proteins from the intra- and extracellular components after CL-6 exposure to atractylodin for 24 and 48 h were involved in cellular (GO:0009987) and metabolic (GO:0008152) processes. The proteins with transporter activity were low in number. Transporter proteins play an essential role in cancer chemotherapy and drug resistance. Modulation of drug transport through decreasing influx and increasing efflux has been shown to be one major mechanism of anticancer drug resistance. Transporter proteins are classified into two types, i.e., the ATP binding cassette (ABC) family and the solute carrier (SLC) family [8]. These transporters function in various pharmacokinetic processes including absorption and excretion of xenobiotics and endogenous molecules. Two transporter proteins identified in this study belong to the ATP-binding cassette sub-family A member 12 (ABCA12), and the ATP-binding cassette sub-family G member 8(ABCG8). The ABCA12 protein is a member of the ATP-binding cassette (ABC) family and it functions as lipid transporters. Up-regulation of ABCA12 was found to be associated with anticancer drug resistance in several types of cancer such as breast [9], ovarian [10], and colorectal cancer [11]. The ATP-binding cassette sub-family G member 8 (ABCG8) belongs to the ATP-binding cassette (ABC) family. This protein is located in enterocyte membranes and plays an essential role in controlling the transportation of dietary cholesterols into and out of the cells. Down-regulation of ABCG8 was observed in breast carcinoma after chemotherapy [9]. Besides, the suppression of ABCG8 induced the apoptosis of liver cancer by inhibiting doxorubicin efflux [12]. Other transporter proteins such as the calcium-binding mitochondrial carrier protein SCaMC-3 (CAD55563) are commonly found in the brain, liver, and small intestine at high density [13]. The function of this protein includes the transportation of adenine nucleotides into mitochondria. Increasing the expression of SCaMC-3 activates the influx of calcium to mitochondria, and thus, induction of cell apoptosis [14]. The transient receptor potential cation channel subfamily V member 6 protein (TRPV6), is a calcium selective cation channel that plays a critical role in calcium uptake in epithelial tissues in epididymis, placenta, and prostate cancer [15,16]. Free intracellular Ca^2+^ activates muscle contraction and cell cycle regulation. In gastric cancer, the increase in free Ca^2+^ in mitochondria via stimulation of TRPV6 protein was shown to result in cancer cell apoptosis [17].

Based on the results of the protein-protein interaction study, atractylodin action was associated with DNA repairing, the cell cycle, and apoptosis. DNA repairing involves several proteins, including breast cancer 2 (BRCA2), BRCA1 associated RING domain 1(BARD1), and proliferating cell nuclear antigen (PCNA). The BRCA2 protein belongs to the tumor suppressor gene family, which is located on chromosome 13. It functions in repairing DNA damage induced by chemical exposure. The PCNA protein is found in yeast, plant, and animal cells. The protein regulates cell division, DNA replication, and cell death through MyD118 and p21 [18]. The induction of cell apoptosis was shown to be associated with a decrease in PCNA expression [19]. BARD1 protein is synthesized in the cell cycle process at the S phase. The function of BARD1 involves DNA damage and repairing, and transcriptional regulation, including apoptosis induction. This protein activates cell apoptosis via p53-phosphorylation [20]. Altogether, results suggest that the mechanism of cytotoxicity of atractylodin is through induction of DNA damage similar to that observed with cisplatin, gemcitabine, and oxaliplatin [21]. This DNA damage results in the activation of cell cycle arrest and apoptosis. The cell cycle process is involved in cell division and is controlled by cell cycle checkpoints at G1, G2, and M checkpoints. Major proteins that are involved in the cell cycle process include cyclin A, cyclin B, cyclin D, cyclin E, cdk2, cdk4, and cdk6. Abnormal functioning of these proteins lead to cell cycle arrest. Atractylodin has previously been shown to induce cholangiocarcinoma cell cycle arrest at the G1 phase [22]. The apoptotic action of atractylodin has also been shown to be both concentration- and time-dependent [22]. Apoptosis, the cell process associated with cell death, involves two main mechanisms, i.e., intrinsic and extrinsic pathways. In the intrinsic pathway, activation of cell apoptosis is through DNA damage. In the extrinsic pathway, on the other hand, activation of cell apoptosis is through the death cell receptor.

The metabolomic study revealed the link between atractylodin action and several metabolic pathways, including aspartate metabolism, pyrimidine metabolism, the citric acid cycle, and ammonia recycling. The primary metabolites identified were glutathione disulfide (GSSG: 65.53-fold change), and taurodeoxycholic acid (TDCA: 22.41-fold change). GSSG consists of two molecules of glutathione linked by a disulfide bond. It plays a role in maintaining redox homeostasis [23]. NOV-002, the glutathione disulfide mimetic, was shown to suppress tumor cell invasion and metastasis via ErbB2, PI3K, Akt, and RhoA [24]. TDCA consists of the bile acid taurine (found in the liver), which is conjugated with deoxycholic acid. This metabolite activates IL-8 gene expression via RelA phosphorylation [25] and induction of HepG2 cell apoptosis [26]. For a better understanding of the atractylodin action, data from both proteomics together with metabolomics were further analyzed by MetaboAnalyst software. Results suggested that the key action of atractylodin is wound healing and blood coagulation. Wound healing is the repair process of skin and tissue following injury. It is an important process that leads to chronic fibrosis, and cancer development/progression. Prolonged and exacerbated healing could trigger chronic fibrosis and tumorigenesis [27]. The blood coagulation process is associated with tumor progression and metastasis [28]. Suppression of this process was previously demonstrated with atractylodin and the colorectal cancer drug avastin [29]. Atractylodin was shown to inhibit wound healing of the cholangiocarcinoma cell CL-6 after 24 and 48 h of exposure [30]. Besides, it also induced the suppression of HO-1 production, STAT1/3 phosphorylation, and NF-κB proteins [30]. The association between atractylodin action and blood coagulation remains to be clarified. The process is induced by two pathways, i.e., intrinsic and extrinsic pathways. The intrinsic pathway is activated by damaging the vessel wall. The factors that control this process are the Hageman factor (XII), plasma thromboplastin (XI), Christmas factor (IX), and Stuart factor (X) [31]. Abnormality of these factors causes hemophilia B and C [32]. The extrinsic pathway is activated by tissue damage outside the blood vessel, which is mainly controlled by the stable factor (VII) [31]. Abnormality of factor VII causes hemophilia A [32].

## 4. Materials and Methods

### 4.1. Cell Culture

The human CCA cell line (CL-6) is the intrahepatic cholangiocarcinoma that was kindly provided by Associate Professor Adisak Wongkajornsilp, Department of Pharmacology, Faculty of Medicine (Siriraj Hospital), Mahidol University. The cell was cultured in RPMI 1640 medium supplemented with 10% (v/v) heated fetal bovine serum (FBS), and 100 IU/mL anti-biotic-antimycotic solution, and maintained at 37 °C, 5% CO_2_ atmosphere, and 95% humidity (HERA CELL 150i, Thermo Scientific, USA).

### 4.2. Sample Preparation

The CL-6 (10^6^ cells) was seeded into a 25 cm^2^ culture flask and incubated overnight at 37 °C, 5% CO_2_ atmosphere. The cell was incubated with atractylodin (40 µg/mL) for 24 h and 48 h. After incubation, proteins and metabolites from intracellular (treated cell) and extracellular components (culture media) were extracted and analyzed by LC-MS/MS.

### 4.3. Proteomics

#### 4.3.1. Protein Extraction

Following exposure of CL-6 cells to atractylodin for 24 h and 48 h, the intra- and extracellular proteins were extracted. For intracellular proteins, the cell was washed with cold PBS and broken using SDS (0.5%). Cell supernatant was separated through centrifugation at 800× *g* (4 °C) for 10 min and stored at −80 °C until use. For extracellular proteins, proteins in the culture media were precipitated by mixing with pre-cold acetone (−20 °C) and incubated overnight at −20 °C. After incubation, the precipitated proteins were separated through centrifugation at 13,000× *g* for 15 min. Protein pellets were dissolved with 0.5% SDS and stored at −80 °C until use. Concentrations of both intra- and extracellular proteins were measured by the Lowry method [33].

#### 4.3.2. SDS-PAGE Analysis and In-Gel Digestion by Trypsin

The proteins (30 µg/sample) extracted from both intra- and extracellular were separated using 12% SDS-PAGE (20 mA constant current for 80 min) and stained using Coomassie Brilliant Blue R. The stained proteins in the gel were cut into 19 small fragments per each sample. The gel fragments were incubated with dithiothreitol (DTT) and iodoacetamide (IAA) before being digested with trypsin and extracted with acetonitrile [34].

#### 4.3.3. LC-MS/MS Analysis

The digested proteins extracted from the gel fragments were dissolved with 0.1% formic acid and injected into LC-MS/MS for protein identification. Briefly, the digested proteins were injected into a μ-pre-column (Monolithic Trap Column, 200 μm i.d. × 5 mm) coupled with the Ultimate 3000 LC system (Dionex), and ESI-Ion Trap MS (HCT ultra PTM Discovery System, BrukerDaltonik) with electrospray at a flow rate of 20 μL/min. Proteins were separated on a nano-column (Monolithic Nano Column, 100 μm i.d.× 5 cm) (Thermo Fisher Scientific, Amsterdam, Netherland)with a solvent gradient mobile phase consisting of solvent A (0.1% formic acid) and solvent B (50% water, 50% acetonitrile, 0.1% formic acid) running at a flow rate of 1 µL/min for 20 min.

#### 4.3.4. Protein Identification and Analysis

The LC-MS/MS data were analyzed using DeCyderTM (Amersham Bioscience AB, Uppsala, Sweden) and MASCOT (http://www.matrixscience.com) programs for protein identification. The PANTHER software (http://www.pantherdb.org) was selected for studying the gene ontology, and the STITCH software was used to study the protein-protein interactions (http://stitch.embl.de).

### 4.4. Metabolomics

#### 4.4.1. Metabolite Extraction

CL-6 cell was cultured in a 25 cm^2^ flask (10^6^ cells/flask) and exposed to atractylodin (40 µg/mL) for 24 h and 48 h. Quenching reagent (cooled methanol) was immediately added (80% v/v of methanol solution) to halt any biochemical reactions. For the metabolite extraction, the samples were divided into two parts, i.e., intracellular metabolites, and extracellular metabolites. For intracellular metabolites, complete media was removed from CL-6 cells after the addition of a quenching reagent. The cell was immediately washed with cold PBS, and transferred to a new tube (1.5 mL) using a cell scraper. Following centrifugation (14,000× *g*, at 4–8 °C for 5 min), the cell supernatant was separated and dried using SpeedVac/lyophilize. These intracellular metabolites were stored at −80 °C until analysis. For extracellular metabolites, the metabolites in media were collected by centrifugation (14,000× *g* for 10 min, 4 °C) and transferred to a new tube (1.5 mL). The extracellular metabolites were dried by SpeedVac/lyophilize and stored at −80 °C until use.

#### 4.4.2. Metabolite Identification

The dried metabolites from both the intra- and extracellular components were reconstituted with LC/MS grade water (500 µL), transferred to a 1.5 mL glass tube, and filtered through the phenix-RC filter (0.2 µm pore size). The metabolites were separated using HPLC (Agilent Technologies, 1200 Infinity Series) and identified by QTRAP^®^ 5500 (AB SCIEX) [35]. The system consisted of security universal HPLC guard cartridge (Phenomenex, CA, USA), amide XBridge HPLC column (3.5 μm × 4.6 mm i.d. × 100 mm length (Waters, MA, USA). The mobile phase consisted of 50% buffer A (95% water, 5% acetonitrile, 20 mM ammonium hydroxide, and 20 mM ammonium acetate adjusted to pH 9.0) and 50% of buffer B (100% acetonitrile). The flow rate was 400 µL/minute with a run time of 23 min. For MS/MS, the spray voltage was set at 3200 V. Nitrogen and argon were used as auxiliary and collision gas, respectively. The scan time for each single-reaction monitoring (SRM) event transition was 0.1 s with a scan width of 1 m/z.

#### 4.4.3. Metabolomics Analysis

The instrument control, data acquisition, and data analysis were achieved using Xcalibar^TM^ software (Thermo Scientific, MA, USA). The up- and down-regulated metabolites were selected for the identification of possible signally pathways using the free online web-based MetaboAnalyst 4.0 (http://old.metaboanalyst.ca/MetaboAnalyst/faces/ModuleView.xhtmL). Moreover, MetaboAnalyst 4.0 was used to determine the correlation between metabolites and proteins in this study.

## 5. Conclusions

In conclusion, by applying proteomic and metabolomic approaches, possible molecular targets of atractylodin in cholangiocarcinoma were shown to be associated with various cell biological processes including the cell cycle, apoptosis, DNA repair, immune response regulation, wound healing, blood vessel development, cell morphogenesis, cell proliferation, Ras protein signal transduction, pyrimidine metabolism, the citrate cycle (TCA cycle), purine metabolism, arginine and proline metabolism, glyoxylate and dicarboxylate metabolism, pentose phosphate pathway, fatty acid biosynthesis, and primary bile acid biosynthesis. Therefore, it was proposed that the action of atractylodin may involve the destruction of DNA of cancer cells, leading to cell cycle arrest and cell apoptosis. The exact molecular targets and their signaling pathways remain to be clarified.

## Figures and Tables

**Figure 1 metabolites-09-00260-f001:**
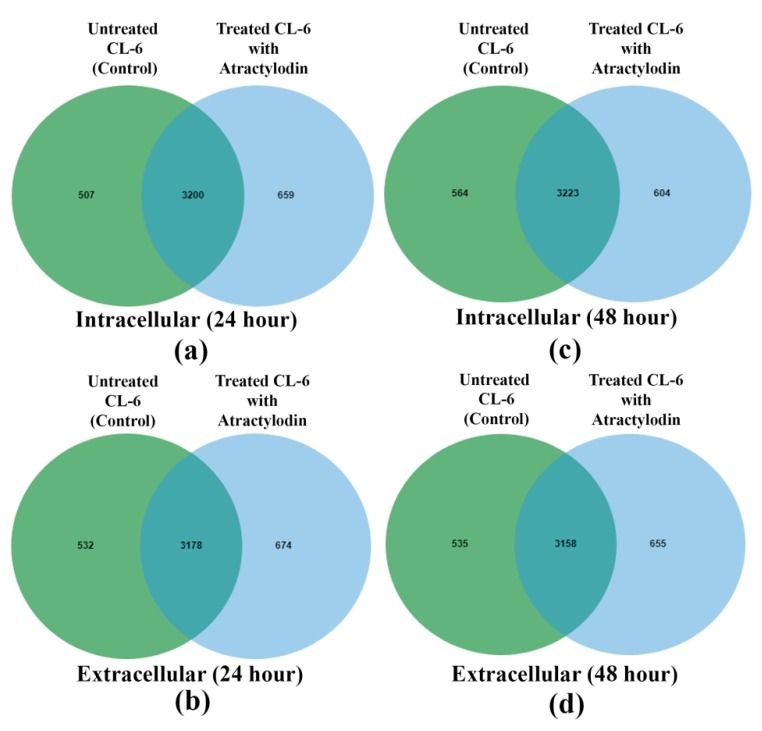
Venn diagrams of intracellular proteins at 24 h (**a**), extracellular proteins at 24 h (**b**), intracellular proteins at 48 h (**c**), and extracellular proteins at 48 h of CL-6 cell incubation (**d**).

**Figure 2 metabolites-09-00260-f002:**
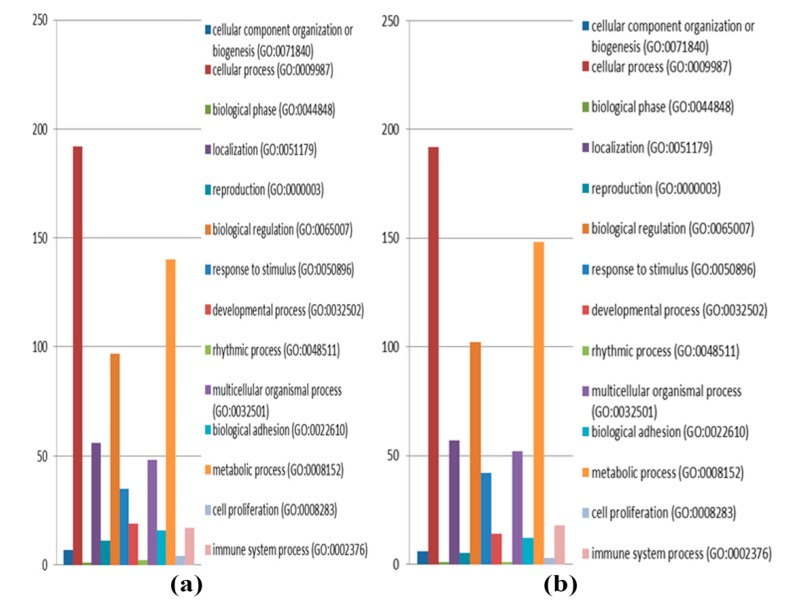
Protein classification based on the biological process of the identified proteins at 24 h (**a**) and 48 h (**b**) of CL-6 cell incubation.

**Figure 3 metabolites-09-00260-f003:**
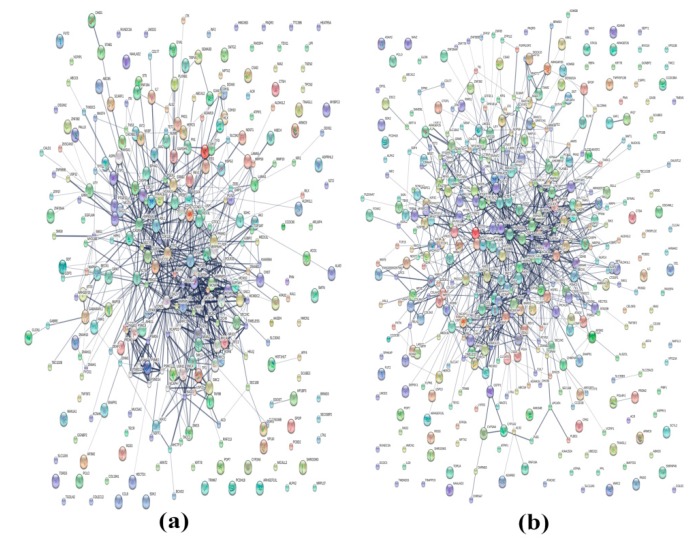
Protein-protein interactions of the identified proteins at 24 h (**a**) and 48 h (**b**) of CL-6 cell incubation by STITCH analysis.

**Figure 4 metabolites-09-00260-f004:**
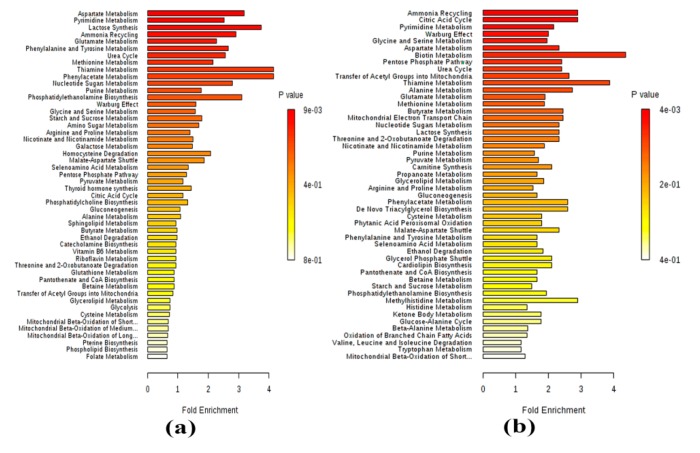
The analysis of metabolites for 24 h (**a**) and 48 h (**b**) of atractylodin exposure by MetaboAnalyst software.

**Figure 5 metabolites-09-00260-f005:**
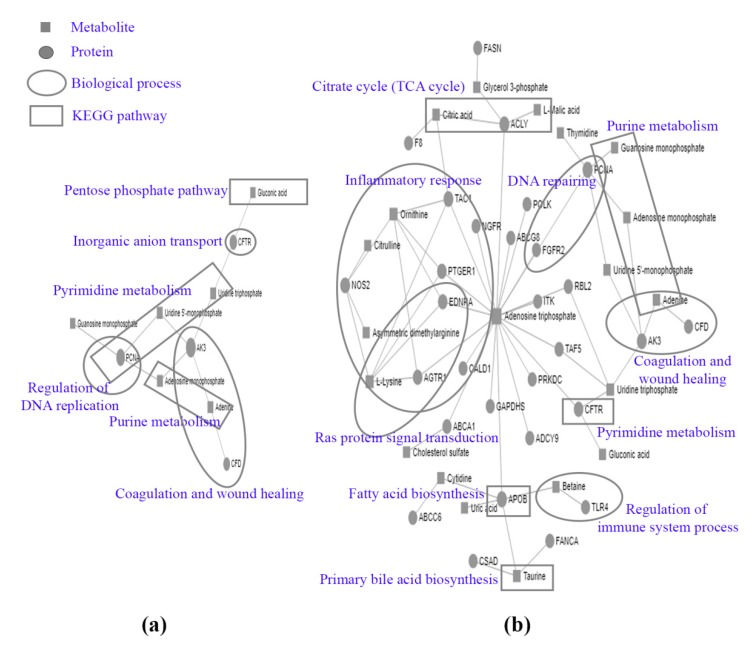
The co-analysis by MetaboAnalyst software of the proteins and metabolites detected at 24 h (**a**) and 48 h (**b**) following atractylodin exposure.

**Table 1 metabolites-09-00260-t001:** Transporter proteins identified at 24 h and 48 h of CL-6 cell incubation.

**Transporter Proteins Identified at 24 h Incubation**		
**Protein Names**	**Protein Names**	**Protein Names**
ATP-binding cassette sub-family A member 12	Polycystic kidney disease protein 1-like 2	Retinal-specific ATP-binding cassette transporter
ATP-binding cassette sub-family G member 8	Polycystic kidney disease protein 1-like 3	Sodium/hydrogen exchanger 9B2
CSC1-like protein 1	Potassium voltage-gated channel subfamily A member 4	Transient receptor potential cation channel subfamily V member 4
Gamma-aminobutyric acid receptor subunit epsilon	Potassium voltage-gated channel subfamily H member 4	Transient receptor potential cation channel subfamily V member 6
Melanoma inhibitory activity protein 2	Potassium voltage-gated channel subfamily H member 8	Voltage-dependent L-type calcium channel subunit beta-2
Mitochondrial inner membrane protein OXA1L	Potassium voltage-gated channel subfamily KQT member 3	V-type proton ATPase 116 kDa subunit α isoform 2
Mitochondrial pyruvate carrier 1	Potassium voltage-gated channel subfamily S member 1	Protein shisa-7
Neuronal acetylcholine receptor subunit alpha-10	Potassium-transporting ATPase subunit beta	
NTF2-related export protein 1	Probable phospholipid-transporting ATPase VD	
**Transporter Proteins Identified at 48 h Incubation**		
**Protein Names**	**Protein Names**	**Protein Names**
Amiloride-sensitive sodium channel subunit gamma	Mitochondrial pyruvate carrier 1	Protein shisa-7
ATP-binding cassette sub-family A member 12	Neuronal acetylcholine receptor subunit alpha-10	Sodium/bile acid cotransporter
ATP-binding cassette sub-family G member 8	Neuronal acetylcholine receptor subunit alpha-7	Sodium/hydrogen exchanger 9B2
Calcium permeable stress-gated cation channel 1	Neuronal acetylcholine receptor subunit beta-3	Synaptic vesicular amine transporter
Calcium-activated chloride channel regulator 4	NTF2-related export protein 1	Transient receptor potential cation channel subfamily V member 6
Calcium-binding mitochondrial carrier protein SCaMC-3	Phospholipid-transporting ATPase IB	Transmembrane channel-like protein 2
Ceramide-1-phosphate transfer protein	Polycystic kidney disease protein 1-like 2	Vacuolar protein sorting-associated protein 26B
Excitatory amino acid transporter 5	Polycystic kidney disease protein 1-like 3	Vesicular acetylcholine transporter
Importin subunit alpha-4	Potassium voltage-gated channel subfamily H member 4	Zinc transporter ZIP6
Melanoma inhibitory activity protein 2	Potassium voltage-gated channel subfamily S member 1	
Mitochondrial glutamate carrier 2	Potassium-transporting ATPase subunit beta	

**Table 2 metabolites-09-00260-t002:** The fold-change of metabolites identified in intra- and extracellular components at 24 h of atractylodin exposure.

**Up-Regulation of Intracellular Metabolites at 24 h of Exposure**					
**Name**	**Fold Change**	**Name**	**Fold Change**	**Name**	**Fold Change**
Inosine	8.05	Threonine	3.37	Taurodeoxycholic acid	2.24
Phenylpyruvate	6.25	S-ribosyl-L-homocysteine	3.31	Phenylalanine	2.22
6-phospho-D-glucono-1,5-lactone	5.86	Indole	2.81	Kynurenine	2.16
Cytidine	5.66	Aconitate	2.73	Folate	2.14
Choline	5.60	Creatine	2.70	Tyrosine	2.13
UDP-D-glucose	5.32	L-arginino-succinate	2.52	2-oxo-4-methylthiobutanoate	2.12
Serine	3.85	D-gluconate	2.24	Pyridoxine	2.01
**Down-Regulation of Intracellular Metabolites at 24 h of Exposure**					
**Name**	**Fold Change**	**Name**	**Fold Change**	**Name**	**Fold Change**
Glutathione disulfide	6.84	AMP	2.71	Cholesteryl sulfate	2.16
Lactate	5.11	Thiamine	2.61	6-phospho-D-gluconate	2.08
Methionine sulfoxide	4.43	Cystine	2.53	Glucosamine	2.06
Glutamine	4.21	Aspartate	2.43	Glucose-1-phosphate	2.04
Citrulline	3.68	Cellobiose	2.41	UTP	2.01
4-aminobutyrate	2.86	Asparagine	2.34		
N-acetyl-glutamate	2.83	dAMP	2.17		
**Up-Regulation of Extracellular Metabolites at 24 h of Exposure**					
**Name**	**Fold Change**	**Name**	**Fold Change**	**Name**	**Fold Change**
Methylnicotinamide	8.57	Glycerate	3.00	Orotidine-5-phosphate	2.12
Adenine	5.81	2,3-dihydroxybenzoic acid	2.73	Serine	2.07
Pyroglutamic acid	3.87	Hydroxyisocaproic acid	2.25	Adenosine	2.04
Cytidine	3.61	Dihydroorotate	2.16		
**Down-Regulation of Extracellular Metabolites at 24 h of Exposure**					
**Name**	**Fold Change**	**Name**	**Fold Change**	**Name**	**Fold Change**
CDP	4.83	N-carbamoyl-L-aspartate	2.79	Acetylphosphate	2.44
Glutathione disulfide	3.96	Orotate	2.71	Citraconic acid	2.20
N6-Acetyl-L-lysine	3.39	2-ketohaxanoic acid	2.66	Glucose-1-phosphate	2.12
Citrate	3.11	2-dehydro-D-gluconate	2.61	Taurodeoxycholic acid	2.11
Phenylpyruvate	3.05	Shikimate-3-phosphate	2.60		
2-oxo-4-methylthiobutanoate	2.84	D-gluconate	2.48		

**Table 3 metabolites-09-00260-t003:** The fold-change of metabolites identified in intra- and extracellular components at 48 h of atractylodin exposure.

**Up-Regulation of Intracellular Metabolites at 48 h of Exposure**
**Name**	**Fold Change**	**Name**	**Fold Change**	**Name**	**Fold Change**
Taurodeoxycholic acid	22.41	Glucosamine	3.09	Taurine	2.25
2-Aminooctanoic acid	21.35	2-oxo-4-methylthiobutanoate	2.97	Citrate	2.23
Cystine	17.21	Allantoate	2.96	Methylnicotinamide	2.20
Orotidine-5-phosphate	6.65	Ribose-phosphate	2.94	N-acetyl-glutamate	2.19
2-dehydro-D-gluconate	5.77	Asparagine	2.89	Ornithine	2.19
Shikimate	5.29	Methylcysteine	2.82	Hexose-phosphate	2.17
Acetylphosphate	4.87	Glycerate	2.67	2,3-dihydroxybenzoic acid	2.14
4-Pyridoxic acid	4.76	N-Acetyl-L-ornithine	2.66	Citrate-isocitrate	2.11
2-ketohaxanoic acid	4.57	Pantothenate	2.63	Acetyllysine	2.11
Shikimate-3-phosphate	4.43	Kynurenic acid	2.56	Thymidine	2.08
Cytidine	4.14	Folate	2.56	Allantoin	2.06
Threonine	4.10	Succinate	2.54	Tryptophan	2.06
Aconitate	4.06	6-phospho-D-gluconate	2.54	Histidine	2.04
N6-Acetyl-L-lysine	3.92	Nicotinamide	2.49	Cholesteryl sulfate	2.04
Cellobiose	3.71	Hydroxyisocaproic acid	2.47	Thymine	2.03
Citrulline	3.45	Glucono-D-lactone	2.41	Orotate	2.02
N-carbamoyl-L-aspartate	3.27	N-acetyl-glutamine	2.40	Deoxyribose-phosphate	2.01
Serine	3.19	CDP	2.31	L-arginino-succinate	2.01
Xanthine	3.14	Lysine	2.30		
Indole	3.10	Phenylpropiolic acid	2.26		
**Down-Regulation of Intracellular Metabolites at 48 h of Exposure**
**Name**	**Fold Change**	**Name**	**Fold Change**	**Name**	**Fold Change**
Glutathione disulfide	61.87	ATP	4.17	IMP	2.78
AMP	6.01	NAD+	4.07	1-Methyladenosine	2.35
Glycerophosphocholine	5.30	GMP	3.62	Kynurenine	2.27
UDP-D-glucose	4.51	UMP	3.25		
**Up-Regulation of Extracellular Metabolites at 48 h of Exposure**
**Name**	**Fold Change**	**Name**	**Fold Change**	**Name**	**Fold Change**
GMP	2.24	Serine	2.94	Adenine	5.51
Glycerate	2.73	UMP	4.12	Cytidine	6.18
S-ribosyl-L-homocysteine	2.77	Methylnicotinamide	4.79		
Down-Regulation of Extracellular Metabolites at 48 h of Exposure
**Name**	**Fold Change**	**Name**	**Fold Change**	**Name**	**Fold Change**
Glutathione disulfide	65.53	UDP-D-glucose	3.63	Allantoin	2.47
CDP	17.44	Phenylpropiolic acid	3.58	Betaine	2.42
6-phospho-D-glucono-1,5-lactone	10.16	Cholesteryl sulfate	3.57	AMP	2.40
2-ketohaxanoic acid	7.30	Uric acid	3.51	Thymine	2.37
Phenylpyruvate	6.62	Methylmalonic acid	3.42	Ng,NG-dimethyl-L-arginine	2.36
Shikimate-3-phosphate	6.57	Deoxyribose-phosphate	3.28	p-hydroxybenzoate	2.33
Citrate	6.33	Kynurenic acid	3.21	Ribose-phosphate	2.29
Citraconic acid	5.77	Hydroxyphenylacetic acid	3.17	Acadesine	2.26
Citrate-isocitrate	5.23	Acetylphosphate	3.14	Hydroxyisocaproic acid	2.26
N-carbamoyl-L-aspartate	5.08	2-oxo-4-methylthiobutanoate	3.01	Biotin	2.24
Octulose-1,8-bisphosphate (OBP)	5.07	Nicotinamide	2.95	SN-glycerol-3-phosphate	2.24
2-dehydro-D-gluconate	5.01	NAD+	2.91	Sarcosine	2.21
D-gluconate	4.79	Shikimate	2.84	Imidazoleacetic acid	2.18
Xanthine	4.11	Purine	2.83	Acetylcarnitine DL	2.11
UTP	4.11	Lactate	2.66	2-Hydroxy-2-methylthiobutanoate	2.10
Allantoate	4.00	Thiamine	2.64	Malate	2.05
L-arginino-succinate	3.96	Acetyllysine	2.58	Pantothenate	2.00
A-ketoglutarate	3.89	1,3-diphopshateglycerate	2.51		
4-Pyridoxic acid	3.86	Thymidine	2.50		
N-Acetyl-L-alanine	3.83	Hexose-phosphate	2.48

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
