# Peer review of "Screening of Molecular Targets of Action of Atractylodin in Cholangiocarcinoma by Applying Proteomic and Metabolomic Approaches"

_metabolites, 2019, doi:10.3390/metabo9110260_

Round 1

Reviewer 1 Report

The present manuscript aimed to investigate the molecular targets of action of actractylodin in a cholangiocarcinoma (CCA) cell line applying proteomics and metabolomics approaches. The Authors found possible molecular targets of atractylodin in CCA associated with apoptosis, DNA repair, Immune response, wound healing, blood vessel development, cell proliferation and additional processes. The effect of the compound may involve the destruction of DNA in cancer cell lines. The paper is really interesting but it is important to clarify some points:

- The Authors utilized the human CCA cell line CL-6, but which type of CCA is? If we refer to the Introduction section, they mentioned the CCA classification in intrahepatic (iCCA), perihilar (pCCA) and distal (dCCA), which is based not only on the anatomical position but it contains also important differences in terms of epidemiology, pathobiology and molecular biology (PMID: 25892683). For that reason, it is crucial to know and to explain better the origin of LC-6 cell line.

- Related to the previous point, it should be very interesting to underline the possible differences in the effect of actractylodin in the several types of CCA. If it is possible, the Authors can try to repeat the same protocol at least in another CCA cell line or they can try to speculate regarding the possible different role of the compound in the subtypes of CCA.

- In the end, the Authors show actractylodin with a key action in the inhibition of wound healing and blood coagulation. In the Discussion section, they should clear up the role of these two processes in cholangiocarcinoma.

Author Response

- The Authors utilized the human CCA cell line CL-6, but which type of CCA is? If we refer to the Introduction section, they mentioned the CCA classification in intrahepatic (iCCA), perihilar (pCCA) and distal (dCCA), which is based not only on the anatomical position but it contains also important differences in terms of epidemiology, pathobiology and molecular biology (PMID: 25892683). For that reason, it is crucial to know and to explain better the origin of LC-6 cell line.

Response: CL-6 cell line is intrahepatic CCA which was obtained from a patient with late-stage CCA from the northeastern region of Thailand. The primary cell was cultured and maintained as a cell line.  The detail of this cell line is now added in the manuscript.

- Related to the previous point, it should be very interesting to underline the possible differences in the effect of actractylodin in several types of CCA. If it is possible, the Authors can try to repeat the same protocol at least in another CCA cell line or they can try to speculate regarding the possible different role of the compound in the subtypes of CCA.

Response:  Thank you for your valuable comment. The study is considered preliminary and will be confirmed in other CCA cell lines focusing on the pathways and targets obtained from this study.

- In the end, the Authors show actractylodin with a critical action in the inhibition of wound healing and blood coagulation. In the Discussion section, they should clear up the role of these two processes in cholangiocarcinoma.

Response:  The role of these processes is now added in the revised manuscript.

Reviewer 2 Report

interesting paper, state of the art approaches, conclusions supported by data

minor:

pathway info supported by metabolomics and proteomics date should be depicted in a separate figure

title: "of" should be added

Author Response

Comments and Suggestions for Authors

The paper is written nicely and adds some interesting information. The methods are adequate as well as the discussion.

 Some points should be modified/added prior to publication.

The title should read: “Screening of molecular targets of action of atractylodin….”

Response:  The title has been revised as suggested.

The legend to Fig. 5 should be modified to be self-explanatory without reading the text, symbols should be explained. A figure should be added combining metabolic pathways with metabolites highlighted matching concomitant protein changes in response to actractylodin treatment

Response: Figure 5 has been revised, combining metabolic pathways with metabolites and proteins. 

Round 2

Reviewer 1 Report

All my points have been answered, in my opinion the paper is can be accepted for publication.